# A Customizable Suite of Methods to Sequence and Annotate Cattle Antibodies

**DOI:** 10.3390/vaccines11061099

**Published:** 2023-06-14

**Authors:** Kristel Ramirez Valdez, Benjamin Nzau, Daniel Dorey-Robinson, Michael Jarman, James Nyagwange, John C. Schwartz, Graham Freimanis, Angela W. Steyn, George M. Warimwe, Liam J. Morrison, William Mwangi, Bryan Charleston, Marie Bonnet-Di Placido, John A. Hammond

**Affiliations:** 1The Pirbright Institute, Pirbright GU24 0NF, UK; 2Roslin Institute, Royal (Dick) School of Veterinary Studies, University of Edinburgh, Midlothian EH25 9RG, UK; 3KEMRI-Wellcome Trust Research Programme CGMRC, Kilifi P.O. Box 230-80108, Kenya

**Keywords:** antibody discovery, IgMAT, 10x Genomics, antibody sequencing, antibody repertoire

## Abstract

Studying the antibody response to infection or vaccination is essential for developing more effective vaccines and therapeutics. Advances in high-throughput antibody sequencing technologies and immunoinformatic tools now allow the fast and comprehensive analysis of antibody repertoires at high resolution in any species. Here, we detail a flexible and customizable suite of methods from flow cytometry, single cell sorting, heavy and light chain amplification to antibody sequencing in cattle. These methods were used successfully, including adaptation to the 10x Genomics platform, to isolate native heavy–light chain pairs. When combined with the Ig-Sequence Multi-Species Annotation Tool, this suite represents a powerful toolkit for studying the cattle antibody response with high resolution and precision. Using three workflows, we processed 84, 96, and 8313 cattle B cells from which we sequenced 24, 31, and 4756 antibody heavy–light chain pairs, respectively. Each method has strengths and limitations in terms of the throughput, timeline, specialist equipment, and cost that are each discussed. Moreover, the principles outlined here can be applied to study antibody responses in other mammalian species.

## 1. Introduction

The specificity and magnitude of the B cell response to vaccination and infection dictates the efficacy of the mammalian humoral response to infectious diseases [1,2,3,4]. Cattle are a global protein source that remain burdened by a range of pathogens that reduce health and productivity, such as foot-and-mouth disease, trypanosome, or bovine respiratory disease [5,6,7,8]. A detailed high-resolution characterization of the antibody response in cattle is essential to help address the need to develop new or improve existing vaccines for often complex pathogens.

Immunoglobulins (Ig) are B cell antigen receptors that can be cell-surface expressed or secreted as circulating antibodies by plasma cells, which produce large quantities of antibodies tailored to specific antigens [9]. Their structure comprises two disulfide-bond-linked heterodimers of a heavy (H) and light (L) chain. Each chain encompasses a constant region encoding the isotype, which can be IgM, or generate IgD, IgG, IgA, or IgE for the H chains and IgL (Lambda) or IgK (Kappa) for the L chains. The antigen binding variable region (V_H_ and V_L_ for the heavy and light chains, respectively) is encoded by V, D, J (heavy chains) and V, J (light chains) gene segments. Both the H and L chains have three hypervariable domains (the complementary determining regions, CDRs) that have been shown to be in contact with antigenic epitopes [10,11,12], of which the CDR3 of the H chain (CDRH3) exhibits the highest levels of variability.

Cattle antibodies have several unusual characteristics compared to most other species. They have relatively few functional V_H_ gene segments (12) that are over 90% identical to one another [13]. The diversity of the circulating repertoire is driven through the combinatorial and imprecise assembly of V(D)J gene segments followed by extensive antigen-independent somatic hypermutation of the variable region to generate a virtually infinite diversity of the cattle antibody repertoire [14]. Furthermore, cattle CDRH3 regions average 26 amino acids in length, substantially longer than in human (average 14 amino acids), mouse (average 11 amino acids), and all other species studied [15]. A subgroup of ultralong CDRH3-containing antibodies (average 61–62 amino acids; [16,17,18]) are unique to cattle, with distinct tertiary structures and potentially distinctive paratopes [19]. Cattle light chain usage for all antibodies is dominated by IgL (95%) over IgK (5%) [20]. These characteristics of cattle V gene segment similarity, extensive somatic mutation, and the wide range of CDRH3 lengths requires some bespoke solutions in order to accurately analyze cattle antibody repertoires and responses [15,21].

Monoclonal antibody discovery has two major applications: first, to understand the immune response in the context of disease or vaccination; second, to generate new tools such as therapeutics or diagnostic reagents. The first two techniques for producing monoclonal antibodies with a predetermined specificity were described in 1975 and 1977 based on hybridoma generation [22] and Epstein–Barr virus infection, respectively [23]. These technologies have been extensively used for many years but can suffer from low efficiency (1–3%) and poor species-specific reagent availability [24,25,26,27]. As a higher throughput alternative, antibodies produced by phage display libraries cannot account for native H and L chain pairing [28]. The next generation of methods for antibody discovery combined single B cell sorting followed by single-cell RT-PCR for natural heavy–light chain cloning and sequencing [29,30,31]. Although robust and practical, this method is low throughput (relative to the whole B cell repertoire), time-consuming, and expensive [32]. Recently, the progress in applying next generation sequencing (NGS) to immunoglobulin repertoires [33] and the introduction of droplet and emulsion nucleic acid amplification technologies have advanced the single B cell techniques into a high-throughput method [34]. Microfluidic platforms can isolate single cells in droplets to either PCR-link H and L chains [35,36,37] or to barcode both chains with the same index [34,38], followed by the bulk PCR amplification and NGS of H and L chains [39,40,41,42,43,44]. Similarly, whole antibody repertoire studies have allowed the analysis of bulk antibody chain dynamics but lack natural H and L chain pairing information.

Recent studies showing that pairing affects the antibody structure, specificity, and avidity have confirmed the importance of natural H and L chain pairing during expression for accurate antigen specificity and avidity analyses [45,46,47]. Additionally, various immunoinformatic tools have been developed to analyze antigen receptor NGS data, mainly in humans and mice [48]. Altogether, the development of these new tools allowed a rapid understanding of the humoral response and antigen-specific antibody discovery in the context of the COVID-19 pandemic, impacting vaccine and therapeutic development [49,50]. These technological advances, combined with the development of new analysis pipelines, reagents, and assays adapted to veterinary immunology research, now allow antibody discovery in veterinary species at a comparable resolution and throughput to that in humans and mice [32,51,52,53,54].

Here, we describe a range of validated methods for cattle single-cell antibody sequencing by combining flow cytometry single B cell sorting and (1) limited-cycle semi-nested PCR, cloning, and Sanger sequencing (low-throughput); (2) limited-cycle semi-nested PCR amplification and indexing, followed by NGS (medium-throughput); or (3) flow cytometry bulk sorting and 10x Chromium Next Gel Bead-in-Emulsion (GEM) processing adapted to cattle antibody sequencing (high-throughput). A cattle antibody sequence analysis using the Ig-Sequence Multi-Species Annotation Tool (IgMAT; [51]), showed no bias between methods. We discuss the flexibility of the workflow, emphasized by the interchangeability of the methods within the flow cytometry, PCR amplification, and sequencing modules.

## 2. Materials and Methods

### 2.1. Cattle Peripheral Mononuclear Cell (PBMC) Preparation

PBMCs from three Holstein–Friesian cattle (one per strategy below, all aged between 1 and 2.5 years) were used. The three animals were vaccinated using a standard regime for dairy cattle in the UK to protect them against leptospirosis, respiratory, and clostridial diseases. The PBMCs were isolated from heparinized blood samples as previously described [55]. Briefly, blood samples were diluted 1:2 in Dulbecco’s phosphate-buffered saline without calcium and magnesium (DPBS, Gibco, Thermo Fisher Scientific, Waltham, MA, USA) and layered on SepMate PBMC isolation tubes (Stemcell Technologies, Vancouver, BC, Canada) containing Histopaque 1.083 g/mL (Sigma-Aldrich, Burlington, MA, USA). The tubes were centrifuged for 30 min at 1200× *g*. The mononuclear cell layer was recovered and treated with ACK lysing buffer (Gibco, Thermo Fisher Scientific, Waltham, MA, USA) to lyse the remaining red blood cells. The recovered mononuclear cells were washed twice with cold DPBS and counted using a TC10 Automated Cell Counter (Bio-Rad, Hercules, CA, USA) as recommended by the manufacturer. The cells were resuspended in horse serum (Sigma-Aldrich, Burlington, MA, USA) containing 10% DMSO (Sigma-Aldrich, Burlington, MA, USA) and cryopreserved until further use.

### 2.2. Cell Staining and Sorting

The antibodies were obtained from the Immunological Toolbox (The Pirbright Institute, Woking, UK, unless stated otherwise (Appendix A) and were labeled with fluorochromes using Lightning coupling kits as recommended by the manufacturer (Expedeon, Abcam, Cambridge, UK). All flow cytometry staining steps were performed in wells from V-bottom 96-well plate for 15 min at room temperature and protected from light followed by a single wash with 200 µL of flow buffer containing 2 mM of EDTA (Invitrogen, Thermo Fisher Scientific, Waltham, MA, USA) and 0.5% horse serum (Gibco, Thermo Fisher Scientific, Waltham, MA, USA) and a final wash with 200 µL of DPBS.

For staining strategy 1, the PBMCs were stained with LIVE/DEAD Fixable Aqua (L/D-Aqua, Invitrogen, Thermo Fisher Scientific, Waltham, MA, USA), anti-cattle CD3-PE (Clone MM1A, Bio-Rad, Hercules, CA, USA), anti-cattle CD8a-PE (Clone CC63), anti-cattle light-chain IgL-APC (Clone IL-A58), and ER Tracker Green (Thermo Fisher Scientific, Waltham, MA, USA). Viable CD3^−^ CD8a^−^ IgL^lo^ ER^hi^ cells were single-cell-sorted into Hard-Shell 96-Well PCR plates (Bio-Rad, Hercules, CA, USA) containing 10 µL of buffer per well (10 mM TRIS pH 8.0 with 1000 units/mL of RNasin Plus RNase Inhibitor (Promega, Fitchburg, WI, USA); Figure 1A). After sorting, the plates were sealed with adhesive foil, spun at 1000 rpm for 1 min at 4 °C, and stored at −80 °C until further use. For staining strategy 2, the cells were stained with LIVE/DEAD Fixable Near-IR (L/D-NIR, Invitrogen, Thermo Fisher Scientific, Waltham, MA, USA), anti-cattle CD8a-FITC (Clone CC63), and anti-cattle IgL-PerCP/Cy5.5 (Clone IL-A58). Viable CD8a^−^ IgL^+^ cells were single-cell sorted as described above (Figure 1B). For staining strategy 3, the cells were stained with LIVE/DEAD Fixable Near-IR (L/D-NIR, Invitrogen, Thermo Fisher Scientific, Waltham, MA, USA), anti-cattle CD40-PerCP/Cy5.5 (clone IL-A158), and anti-cattle IgL-DyLight 405 (Clone IL-A58). Live CD40^+^ IgL^+^ cells were sorted into a 1.5 mL microtube (Eppendorf, Hamburg, Germany) containing DPBS with 0.04% BSA (Invitrogen, Thermo Fisher Scientific, Waltham, MA, USA) and immediately used for downstream steps (Figure 1C).

All sorting methods were performed on a BD FACS Aria III Cell Sorter (BD Biosciences, Franklin Lakes, NJ, USA) as described previously [56].

### 2.3. Primer Design

All primers reported here were designed as part of this study, except for the Igλ inner-reverse primer [31] and Read 1/P5+Read 1 primers [57]. To design antibody heavy and light chain constant domain-specific primers, mRNA, genomic, and predicted gene sequences were mined and aligned from public databases and supported by the available scaffolds in the UMD_3.1 and Btau_4.0 genome assemblies (accessions GCF_000003055.4 and GCF_000003205.2, respectively) and publicly available coding sequences (GenBank accessions AY221099, AY221098, AF097213, S82409, S82407.1, AF109167, AF005274, AF515672, AF411240, BC147881, BC146168, BC114850, BC114809, DQ497627, DQ497628, DQ452014, X62916, AY158087, X16702, and X16701) [58,59]. Several primer sites that did not contain any polymorphisms were identified to amplify cattle IgG (3), IgM (2), IgM and IgD (2), IgL (3), and IgK (3). IgK primers were manually designed using a sequence from the cattle reference genome assembly, ARS-UCD1.2 (accession GCF_002263795.1) to cover all functional *IGKV* gene segments. The primer sequences were then verified using the NCBI primer design tool (https://www.ncbi.nlm.nih.gov/tools/primer-blast/ (accessed on 4 May 2020).

Each candidate IgH and IgL constant region primer was used as the gene-specific primer for the 5′ RACE amplification (SMARTer RACE, Takara Bio Inc., Shiga, Japan) of Ig transcripts from four unrelated Holstein–Friesian cattle cDNA derived from PBMCs, as described previously [60]. All successful amplicons were sequenced at the Centre for Genomic Research (The University of Liverpool, Liverpool, UK) using 454 Titanium technology. Contigs were generated using the MIRA assembler [61] with accurate 454 settings. The contigs were viewed and edited using the gap4 and gap5 programs from the Staden packages [62] to generate the final alignment of unique sequences and their overall frequencies. The reverse primers were further validated using more recent genome assemblies to confirm their specificity across at least *Bos* sp. The genomic regions containing the *IGH*, *IGL*, and *IGK* gene loci within the Hereford reference genome (ARS-UCD1.2, accession GCF_002263795.1) and the long-read assemblies for a yak cow (*Bos grunniens*; ARS_UNL_BGru_maternal_1.0_p; accession GCA_009493645.1), Highland bull (*Bos taurus*; ARS_UNL_Btau-highland_paternal_1.0_alt; accession GCA_009493655.1), and water buffalo (*Bubalus bubalis*; UOA_WB_1; accession GCA_003121395.1) were queried for primer sequence conservation using both Artemis [63] and BLAST [64].

All forward V gene primers matched each of the assemblies to varying extents; however, as the V regions are highly repetitive, they are often both poorly annotated and prone to mis-assembly. Combined with this potential for mis-assembly and haplotype variation in the gene content, there is uncertainty in the correct number of V genes present in any given assembly or individual. Therefore, it is difficult to ascertain whether the forward primers enable efficient amplification across bovine species. For this reason, multiple degenerate forward primers based on the available public sequences were used to maximize the number of amplified V genes. Comprehensive heavy and light chain V gene primer sets were designed manually, including some redundancy, to amplify all recovered transcripts. Furthermore, the coverage of each constant region primer was compared to limit bias, with the final primers selected based on complete representation across pools. These V and constant gene primer combinations were then used on the same cDNA samples for the PCR (95 °C for 15 min, 35 cycles as follows: 94 °C for 30 s, 58 °C for 30 s, 72 °C for 45 s, and a final extension at 72 °C for 10 min, using HotStart Taq DNA Polymerase; Qiagen, Venlo, The Netherlands) and sequenced using the same 454 process to validate the comprehensiveness of the V gene amplification. These primers (see the list of primers in Table 1) were then taken forward into the subsequent amplification pipelines.

IgG, IgM, IgL, and IgK primer pairs were validated by the amplification of cattle cDNA using the above-mentioned PCR conditions (see the combination of primers in the Results section). For the next-generation sequencing (medium-throughput strategy), the constant region inner primers used in PCR 3 ensured similar sizes for all isotype amplicons to limit sequencing bias against longer amplicons. For the next-generation sequencing, Illumina overhang adapter sequences were added to the gene-specific primers based on the 16S Metagenomic Sequencing Library Preparation Protocol [65] as an anchor for the indexing primers [66].

### 2.4. Immunoglobulin Gene Amplification and Library Preparation

#### 2.4.1. Strategy 1: PCR Amplification and Cloning Method Used in the Low-Throughput Workflow

A semi-nested PCR was used to amplify the antibody H and L chain variable regions (V_H_ and V_L_, respectively) from sorted single cells. All reactions measured 25 μL, and all primers are described in Table 1. First, a multiplex RT-PCR reaction was performed using the OneStep RT-PCR Kit (Qiagen, Venlo, The Netherlands) as recommended by the manufacturer with 400 μM of dNTPs, 100 μM of forward primers annealed to the 5′ of the variable region, and 100 μM of reverse primers at the 5′ of the constant region of IgG, IgM, or IgL. The RT-PCR reaction mix was incubated at 50 °C for 30 min for cDNA synthesis, followed by an activation step at 95 °C for 15 min and 15 PCR amplification cycles (94 °C for 30 s, 58 °C for 30 s, 72 °C for 1 min), with a final extension step at 72 °C for 10 min (Appendix A). The amplified products were diluted 1:20 and 3 µL was used as a template for further PCR reactions (PCR 2). The PCR 2 reactions were performed using 0.025 U/μL (0.625 U) of HotStart Taq DNA Polymerase (Qiagen, Venlo, The Netherlands) with 200 µM of dNTPs and 100 µM of each primer. The amplification procedure was as follows: 95 °C for 15 min, followed by 49 cycles (94 °C for 30 s, 58 °C for 30 s, 72 °C for 1 min) and a final extension step of 10 min at 72 °C (Appendix A). The PCR products were resolved on a 2% low melting temperature agarose gel (Sigma-Aldrich, Burlington, MA, USA) at 100 volts for 45 min and the amplicons (of approximately 500 bp for V_H_ and 400 bp for V_L_) were excised and purified using the Monarch DNA Gel Extraction Kit (New England Biolab, NEB, Ipswich, MA, USA), cloned into pGEM-T Easy Vector (Promega, Fitchburg, WI, USA), and propagated in *E. coli* JM109 competent cells (Promega, Fitchburg, WI, USA). The selection was performed using LB agar media plates containing 100 μg/mL of ampicillin (Sigma-Aldrich, Burlington, MA, USA) and supplemented post-solidification with 100 μL of 100 mM IPTG (Sigma-Aldrich, Burlington, MA, USA) and 20 µL of 50 mg/mL X-Gal (5-bromo-4-chloro-3-indolyl-β-D-galactopyranoside, Promega, Fitchburg, WI, USA). White colonies were picked and the correct insert was confirmed by PCR using M13 Forward and Reverse Primers [67] and DreamTaq DNA Polymerase (Thermo Fisher Scientific, Waltham, MA, USA).

#### 2.4.2. Strategy 2: PCR Amplification and Indexing Method Used in the Medium-Throughput Workflow

Here, cDNA synthesis and single-plex semi-nested PCR reactions were used to amplify the antibody H and L chains. First, the cDNA was synthesized from sorted single cells as previously described [56] in a 20 μL reaction containing 1 μL of Sensiscript reverse transcriptase (Qiagen, Venlo, The Netherlands), 2 µM dT23VN oligonucleotides (NEB, Ipswich, MA, USA), 4 µM Random Primer 9 (NEB, Ipswich, MA, USA), 10 units of RNaseOUT (Thermo Fisher Scientific, Waltham, MA, USA), and 500 μM of dNTPs (Qiagen, Venlo, The Netherlands), and incubated at 37 °C for one hour. The amplification of the V_H_ and V_L_ was achieved in 25 μL reactions using a semi-nested PCR approach, using 4 μL of the cDNA product and 0.016 U/μL (0.4 Units) of Q5 Hot Start High-Fidelity DNA Polymerase (NEB, Ipswich, MA, USA), with 200 µM of dNTPs, 50 µM of each primer, Q5 high-GC enhancer as recommended by the manufacturer, and 1 M of betaine (Sigma-Aldrich, Burlington, MA, USA). All primers are described in Table 1. Touch-down PCR 1 was composed of 30 s of activation at 98 °C, followed by five cycles at 63 °C (98 °C for 10 s, 63 °C for 15 s, and 72 °C for 20 s), then 20 cycles at 58 °C (98 °C for 10 s, 58 °C for 15 s, and 72 °C for 20 s) and a final extension step of 2 min at 72 °C (Appendix A). The PCR 1 products were used as the template for a second round of PCR reactions (named PCR 3) in which overhang (-OH) Illumina adapter sequences were added to primers as anchors for the indexing PCR [65,68]. The PCR 3 reaction mix was as above and the amplification was performed in two steps starting with five cycles at a lower annealing temperature (58 °C) followed by either 30, 25, 27, and 28 cycles at 63 °C annealing for IgG, IgM, IgL, and IgK, respectively (Appendix A). The expected amplicon sizes were approximately 600 bp. The PCR products were purified with AMPure XP beads (Beckman Coulter Life Sciences; Indianapolis, IN, USA) using a 0.8 bead/amplicon volume ratio following the manufacturer’s protocol. MiSeq Illumina indexes [66] were added using an 8 cycles amplification PCR (PCR 4: 98 °C for 10 s, 58 °C for 15 s, 72 °C for 20 s; [68]). The final products were purified using the AMPure XP beads.

#### 2.4.3. Strategy 3: PCR Amplification Method Used in the High-Throughput Workflow

Sorted B cells were processed on a 10x Chromium Controller device using Next GEM Single Cell V(D)J v1.1 Reagent Kits (10x Genomics, Pleasanton, CA, USA) following the manufacturer’s instructions. Briefly, the cells were prepared for loading in 1× PBS with 0.04% *w*/*v* BSA and counted on a TC10 Automated Cell Counter (Bio-Rad, Hercules, CA, USA). The cells were split into three samples containing approximately 1500, 3000, and 7700 cells for recovery of approximately 900, 1800, and 4500 antibodies (designated samples 1, 2, and 3, respectively) and dispensed into three channels of a G-chip to generate single-cell Gel Bead-in-Emulsions (GEMs). The formed GEMS were collected into 0.2 mL microtubes (Eppendorf, Hamburg, Germany) and a GEM reverse transcription reaction to generate cDNA was performed at 53 °C for 45 min, followed by inactivation for 5 min at 85 °C. A GEM clean-up was performed using Dynabeads MyOne SILANE (10x Genomics, Pleasanton, CA, USA) followed by amplifying barcoded full-length cDNA from poly-adenylated mRNA with 16, 14, and 13 cycles for 900, 1800, and 4500 target cell recovery, respectively. Sample 3 cDNA was used to test the compatibility of three constant reverse primers of each isotype (IgM_C1-3, IgG_C1-3, IgL_C2, IgL_C4, Ig λ chain inner [31], IgK_C1-3) with 10x forward primer (Read 1) [57] using Kapa HiFi Taq polymerase (Roche, Basel, Switzerland) for 3 min at 95 °C, followed by 32 amplification cycles (98° C for 20 s, 62 °C for 15 s, and 72 °C for 30 s) and a final extension step of 2 min at 72 °C. Two reverse primers were selected for each isotype to process samples for NGS (see Table 1). The first semi-nested V(D)J target enrichment was performed using Read1 and constant region-specific outer reverse primers. In contrast, the second semi-nested PCR used the P5+Read1 forward primer with constant region inner primers (Table 1). Two trials with different PCR conditions were tested. Trial 1 consisted of 6 cycles in the first PCR reactions and 8 cycles in the second PCR reactions as recommended by 10x Genomics protocols. Trial 2 consisted of two additional cycles for the first PCR (8 cycles in the first PCR) followed by four extra cycles for the second PCR (12 cycles; Appendix A). The PCR amplicons were profiled using a Bioanalyzer High-Sensitivity DNA Kit (Agilent Technologies, Inc., Santa Clara, CA, USA) followed by library preparation and indexing for multiplexing according to the manufacturer’s instructions. The libraries were bead-purified and profiled again by the bioanalyzer before sequencing.

### 2.5. Sequencing

For the low-throughput strategy, two clones bearing inserts at expected sizes (500 bp for H and 400 bp for L) were Sanger sequenced for each sample and chain (ABI 3730XL, Eurofins Scientific, Luxembourg, Luxembourg). Below 6 nucleotides differences between clones (calculated using Taq polymerase error rate as 1.8 × 10^−4^ errors/base/doubling and the number of PCR cycles used; [69]), sequences were considered to arise from the same antibody transcript.

For the medium-throughput method, amplicons derived from individual samples were visualized on a Tapestation 4200 using a DNA D1000 ScreenTape (Agilent Technologies, Inc., Santa Clara, CA, USA). Samples exhibiting a band at approximately 600 bp were pooled at a 5 nM final concentration. The library pool was quantitated using the NEBNext Library Quantitation Kit for Illumina (NEB, Ipswich, MA, USA) and confirmed using the Qubit fluorometer (Thermo Fisher Scientific, Waltham, MA, USA). The sequencing pool was loaded on a MiSeq v3 reagent kit (Illumina Inc., San Diego, CA, USA) for 2 × 300 cycle paired-end sequencing according to the manufacturer’s specifications with a 40% PhiX spike-in to introduce diversity.

For the high-throughput strategy, pooling was performed following the 10x Genomics pooling worksheet to produce a target library pool concentration of 5 nM, aiming to achieve a target of 5000 reads per cell (5000 reads/cell). The final library pool was quantitated using the NEBNext Library Quantitation Kit for Illumina (NEB, Ipswich, MA, USA) and confirmed using the Qubit fluorometer (Thermo Fisher Scientific, Waltham, MA, USA). The sequencing pool was loaded on a NextSeq 150 Cycle High-Output Cartridge (Illumina Inc., San Diego, CA, USA) for paired-end sequencing on a NextSeq 550 sequencer according to the manufacturer’s specifications. Samples from trial 1 were sequenced once, while samples from trial 2 were sequenced twice.

### 2.6. Bioinformatic Analysis

Alignments of sequences obtained from the low-throughput method were created using MEGA version 7 software [70], and gene usage data were generated using IgBLASTn (v1.0.0, https://pubmed.ncbi.nlm.nih.gov/23671333/ (accessed on 22 November 2021)) with V, D, and J sequences from IMGT [71]. The region annotation was performed by translation into six reading frames using BBMAP translate6frames (V38.71, https://jgi.doe.gov/data-and-tools/bbtools/bb-tools-user-guide/bbmap-guide/ (accessed on 22 November 2021)). Sequences that did not contain stop codons were sequence-annotated with IgMAT [51] using hidden Markov models built from IMGT cattle data. For the medium-throughput method, where a dual combinatorial indexing strategy was used, demultiplexing was carried out on the Illumina sequencer. Subsequently, each sample was quality trimmed with Trim Galore (v0.5.0, https://github.com/FelixKrueger/TrimGalore (accessed on 24 November 2021)) using a quality cut-off of 30 before the reads were merged using FLASH (v1.2.11; [72]). The gene usage and region annotation was performed with IgBLASTn, BBMAP translate6frames, and IgMAT as above. Full amino acid sequences were further clustered together using an amino acid identity rate of 96% using UCLUST (v11.0.667; [73]) and the top 10 clusters were extracted for further processing. Single chains per cell were determined by dominant clusters encompassing over 90% of the UCLUST input sequences. Where multiple dominant clusters were present, each dominant clusters’ centroid was assigned as that sample’s amino acid sequence.

For the high-throughput method, samples from independent channels were demultiplexed using cellranger mkfastq (10x Genomics Cell Ranger v6.1.1, https://support.10xgenomics.com/single cell-vdj/software/downloads/latest (accessed on 1 December 2021)). Cellranger vdj was ran using inner primers from the semi-nested PCR and cattle reference created from the IMGT cattle antibody sequences using the 10x script fetch-imgt following the developers’ instructions (https://support.10xgenomics.com/single cell-vdj/software/pipelines/latest/advanced/references (accessed on 1 December 2021); Appendix A). The gene usage and region annotation was performed with IgBLASTn, BBMAP translate6frames, and IgMAT as above.

## 3. Results and Discussion

### 3.1. Flow Cytometry

We have developed three broad strategies for cattle B cell selection by flow cytometry (Figure 1). In the first approach, we aimed to enrich antibody-secreting plasma cells, which are known to have low levels of antibodies on their cell surfaces [74] and high endoplasmic reticulum activity [75,76]. To target the plasma cells, we stained and excluded dead cells, gated on singlets using FSC-A and FSC-H parameters, and used anti-cattle CD3 and CD8a surface markers to exclude T cells. An antibody for surface IgL combined with ER staining selected the population of IgL-low (IgL^lo^) cells with a high ER expression signal (ER^hi^). IgL^lo^ ER^hi^ cells comprised 0.76% of the live CD8a^−^ CD3^−^ singlet cells (Figure 2A) and 1.78 × 10^−3^% of live singlet PBMCs, which is broadly equivalent to what has been described from plasma cells in humans and mice [76].

Vaccine efficacy prediction may also require the analysis of the overall B cell population (i.e., including memory cells) rather than plasma cells alone [77]. For the second staining strategy, we gated on the 27.2% of IgL^+^ cells from live CD8a^−^ singlet PBMCs (Figure 2B) and sorted IgL^+^ single cells. Comparing the above staining strategies, plotting IgL against CD3 or CD8a expectedly showed no overlap and we concluded that staining for CD3 and CD8a was superfluous, in line with cattle B cell phenotyping strategies [53]. However, to provide more confidence in cell phenotyping, we added CD40 to the panel (a marker for sustained antibody production in humans and mice [78]). In this third strategy, CD40 enabled gating on B cells before selecting the 16.8% of IgL^+^ cells (Figure 2C). Approximately 12,000 sorted CD40^+^IgL^+^ cells were immediately processed on the 10x Chromium Controller for antibody V_H_ and V_L_ region amplification and sequencing.

These three simple flow cytometry strategies based on IgL surface staining isolate overlapping B cell phenotypes that can be preferentially selected depending on the research question and available reagents, independent of any downstream processing. Additional markers such as CD20, CD21, CD71, or CD138 can easily be included to select other and more specific B cell subsets [53]. As vaccine efficacy can be correlated to the antigen-specific antibody response [79,80,81,82], fluorescently labeled antigens could also be added to the flow cytometry panel for sorting antigen-specific B cells [21,83] for downstream antigen-specific antibody discovery [31].

### 3.2. Primer Design and Validation

Cattle exist as many breeds and subspecies due to their history of domestication. Over the last decade, the number, and importantly the quality, of cattle genomes has increased, allowing significantly more confidence in the design of PCR primers to conserved regions or within immunogenetic regions. However, the success of bulk and single-cell PCR strategies for antibody discovery across multiple gene segments and Ig subclasses relies on primer design, ensuring that they cover all gene segments, all SNPs are considered, and a good level of efficiency is achieved.

V_H_ and V_L_ were amplified using isotype-specific reverse constant region primers by 5′ RACE followed by 454 sequencing to enable the design of V gene segment primers. Between 24,000 and 36,000 reads per amplicon were recovered. As predicted, the amplification with IgK primers did not yield enough product for sequencing and was not pursued further using this strategy. Forward degenerate primers designed to anneal to all functional heavy and light chain V gene segments were used to amplify the cDNA and sequenced using the same 454 sequencing process to assess the comprehensiveness of the V gene segment amplification. The sequence recovery in the most abundant 500 transcripts was remarkably similar, with over 98% of sequences being in both the 5′ RACE and PCR amplicon pools, and the most abundant 20 sequences from one all ranked in with the top 22 from the other.

The reverse isotype-specific constant region primers were examined for their sequence conservation using a sampling of diverse long-read bovine genome assemblies (i.e., water buffalo, yak, and Highland cattle). All constant-region specific primers were overall highly conserved across bovines, with the exception of IgM_C1 and IgL_C2 (see details in Appendix A). For all antibody chains, the designed constant region reverse primers were tested independently with one V primer and showed amplification at the expected size (Figure 3A, first row). All V primers were tested independently with a constant region primer for each isotype to confirm its efficiency (Figure 3A). Band intensity variations may be explained by the number of gene segments covered by each degenerate primer or the frequency of usage of the associated (set of) gene segments. We were able to design primers to amplify all known functional V gene segments at expected sizes (400–600 bp; Figure 3) for IgG, IgM, and IgL. For the singleplex amplification used in strategy 2, we included IgK primers. Of note, the IGKV1.1 primer failed to amplify transcripts (sample 20), which correlates with the classification of the associated V gene segment as an ORF [71]. The designed primers ensured all amplicons used in the second round of PCR amplicons were similar enough in size to avoid NGS sequencing bias against longer amplicons [84]. To establish the 10x Chromium, cDNA generated from GEMs was used to test several constant region reverse primers for each isotype combined with the Read1 10x primer. Figure 3B shows that all constant region forward primers tested were compatible. For each isotype, we selected two efficient reverse primers for the first and the semi-nested PCR of the V(D)J amplification. Altogether, we designed and validated a set of primers for each isotype and strategy for cattle antibody amplification, adaptable to downstream sequencing workflows (Table 1).

### 3.3. PCR Strategy and Sequence Analysis

After the flow cytometric sorting of B cells, we explored three antibody chain amplification strategies (Figure 1). The sequencing strategy determines the ultimate throughput of the antibody sequencing, while upstream isolation is more flexible and adaptable to the project needs (See graphical abstract).

#### 3.3.1. Strategy 1: Antibody Amplification and Sequencing in a Low-Throughput Workflow

The first strategy for antibody VH and VL amplification and sequencing was based on a single-step multiplexed RT-PCR, in which cDNA synthesis was followed by simultaneous amplification of the heavy (IgM or IgG) and light (IgL) chains. IgK, representing only 5% of cattle light chains [20], was not included due to multiplexing difficulties and a general lack of expression. A semi-nested PCR (PCR 2) was subsequently performed with constant region inner primers. Next, gel-purified amplicons of wells exhibiting PCR bands for both H and L chains were cloned into a pGEMTeasy vector, Sanger-sequenced, and analyzed using IgBLASTn and IgMAT [51]. We PCR-amplified H and L chains from 39.3% and 48.8% of the wells (Figure 4A and Appendix A). Out of 84 sorted cells we were able to PCR an amplicon of the expected size for both H and L chains from 26 wells, of which 24 (28.6%) consisted of a single H and L chain after sequencing (Figure 4A and Appendix A). The few wells that produced more than one H or L chain likely contained more than one cell from the original sorting or potentially a level of biallelism [36,85]. We were only able to amplify an unpaired H or L chain in 7 and 15 wells, respectively. Overall, 42.9% of the wells produced no product, which could be a consequence of many factors, including (i) the sorting parameters and gating strategy resulting in the absence of cells, the sorting of non-B cells, or cells falling on the walls of the wells; (ii) a primer design that did not account for all SNPs (which we consider unlikely due to the primer design strategy); (iii) sample processing that might have degraded the RNA, such as the time between sorting and freezing; or (iv) the number of PCR cycles did not permit the amplification of rare transcripts.

#### 3.3.2. Strategy 2: Antibody Amplification and Sequencing in a Medium-Throughput Workflow

For our second approach, we combined a semi-nested PCR with the Illumina 16S Metagenomics library preparation protocol that ensures unique combinatorial dual indexing of amplicons for subsequent NGS sequencing and demultiplexing [65]. We also included the IgK chain but excluded the V_K_L2_ primer, which was inhibiting the IgK PCR efficiency (Appendix A). We synthesized cDNA in the first step followed by four independent PCR 1 runs for IgM, IgG, IgL, and IgK chains. Primers in the semi-nested PCR ensured that the V_H_ and V_L_ amplicons were of similar sizes to avoid NGS sequencing bias for smaller amplicons. They encompassed Illumina adapters used as anchors for the indexing PCR. H and L amplicons from the same well were indexed with identical index primers to facilitate demultiplexing. The final amplicons were loaded on a Tapestation (Agilent Technologies, Inc., Santa Clara, CA, USA) and wells yielding a band at 600 bp were pooled at equimolar concentrations for Illumina sequencing. Due to similarities between amplicons, 40% of PhiX was spiked in to increase the library diversity. Sequences were then analyzed using a custom pipeline and IgMAT. Out of the 96 sorted cells, we were able to amplify and sequence 31 H-L antibody pairs (32.3%); 27 wells (28.1%) did not produce any amplicon, while we obtained more than one heavy, light, or both chains in 8, 7, and 3 of the wells, respectively (18.8%). In total, we amplified 64.6% of the heavy chains and 58.3% of the light chains (Figure 4A, Appendix A). Altogether, these ranges of H-L chain recovery are comparable for both low- (28.6%) and medium- (32.3%) throughput methods.

A similar medium-throughput approach using a dual indexing strategy can also be used to bulk sequence whole repertoires of circulating B cells by indexing samples from distinct animals, time points, and tissues (as opposed to indexing single cells). This would allow the tracking of antibody sequences from samples taken at different points during infection or vaccination studies [80] from several individuals and in multiple tissues, all in one sequencing run. With the possibility to pool up to 384 samples (considering the number of currently available Illumina indexes), the limiting factor is the sequencing depth.

#### 3.3.3. Strategy 3: Antibody Amplification and Sequencing in a High-Throughput Workflow

To further increase the throughput of the paired H-L chain sequencing, we adapted the 10x Genomics workflow [57] for antibody discovery to cattle by using droplet-based partitioning with the 10x Chromium Controller combined with the amplification of V_H_ and V_L_. We aimed to recover 900, 1800, and 4500 cells in three samples (1, 2, and 3, respectively). The PCR target enrichment conditions recommended by 10x Genomics were used in trial 1, while trial 2 had additional cycles in both PCRs to evaluate the effect on H-L pair recovery, in line with multiple studies demonstrating that increasing the cycle number improves the overall performance [86,87]. The data were processed using Cell Ranger with default settings. All V(D)J processed single-cell samples exhibited Q30 scores above 90% and largely exceeded the recommended number of reads/samples, with an average of 17,000 reads/cell. The Cell Ranger estimates of sequenced cell numbers were 596 and 663 cells for sample 1, 1639 and 1683 cells for sample 2, and 2966 and 4581 cells for sample 3 of trial 1 and trial 2, respectively. The differences between trial 1 and trial 2’s estimated numbers of cells may have been related to the increased amplification of transcripts that pass filtering thresholds or the sequencing depth. As we obtained similar numbers of H-L pairs for the two trials of each sample (Figure 4B and Appendix A–E), we concluded that increasing the PCR cycle number did not increase the recovery of paired antibody H-L chains.

The average ratios of GEMs exhibiting more than one H or L chain ranged from 5.21% for sample 1 to 5.93% for sample 2 and 6.25% for sample 3, and the benefit of increasing the cell numbers per channel was supported by the higher number of antibody sequences recovered (Appendix A). With both trials and sequencing runs combined, we obtained 496 (sample 1), 1390 (sample 2), and 2870 (sample 3) single H-L pairs, for a total of 4756 antibody H-L sequences (Appendix A), dramatically increasing the throughput compared to techniques based on single-cell sorting in 96-well plates, as expected. To our knowledge this is the first experiment using 10x technology to sequence cattle antibodies. The 5′ RACE-based 10x protocol ensures no bias against V gene segment usage. We aligned all V_L_ reads to the V_L_12set of primers used for the low-throughput strategy, confirming that 98% of the lambda chain reads were covered by V_L_12set exact sequences, with 99.6% allowing 1 mismatch. This validates the completeness of the V_L_12set primers. A similar analysis was not possible for the V_H_, V_L_5, and V_K_ sets of primers, as the primer sequence is located on the leader sequence trimmed by Cell Ranger.

### 3.4. Cattle Antibody Characterization

The IgMAT based annotation of antibody sequences isolated in this study confirmed the expected structure of the framework and complementary determining regions, from which we could clearly identify the CDR3. The overall distribution of CDRH3 lengths was in line with previous studies, with average lengths of 20–24 amino acids for the three methods ([51,88]; Figure 5A, Appendix A). Overall, 53 ultralong heavy chain sequences (CDRH3 length ≥ 50 amino acids) paired with light chains were amplified at 4.2% (1 antibody), 3.2% (1 antibody), and 1.1% (51 antibodies) for the low-, medium-, and high-throughput approaches, respectively (Appendix A). These percentages were lower than the previously described 5–10% [89,90], which could be attributed to (i) PCR bias against ultralong antibodies, (ii) NGS sequencing bias against longer amplicons, or (iii) variation in ultralong heavy-chain proportions depending on individuals.

Given that cattle VH gene segments share an average of 90% identity [13], the genomic origin of a single antibody sequence is often ambiguous depending on the level of somatic mutation. Therefore, the most probable V gene segment used was assigned to each sequence, and when ambiguous alternative V gene segments were added (Appendix A). All VH gene segments belonged to the IGHV1 gene family, and all of the ultralong heavy chains were identified as IGHV1-7 and IGHD8-2, as previously described [16], except one that aligns to the IGHV1-20 gene segment. We also observed restricted pairing of 86.8% (46) of the ultralong heavy chains to light chains using the VL1-47 gene segment, out of which 35 were the so-called invariant light chain CDRL3 previously described (ASAEDSSSNAV; [18]), with another seven paired light chains exhibiting a single amino acid difference to the conserved CDRL3 (Appendix A). Overall, we did not observe significant differences in CDRH3 length distribution (Figure 5A) or gene usage (Appendix A) between the three approaches, implying no obvious bias between the three amplification strategies.

IgM-expressing cells dominated, with IgG comprising only 30.6% of cells processed in strategy 1, 2.7% for strategy 2, and 5.8% for strategy 3 (Figure 5B, Appendix A). These data indicate a potentially higher frequency of IgG-secreting cells within the IgL^lo^ ER^hi^ population compared to the overall B cell population, which are likely to be plasma cells and plasmablasts. From the mid-throughput experiment, three light chains were confirmed to be IgK, corresponding to 4.5% of all amplified light chains (including from multiplets), in line with the described 5% of the repertoire ([20]; Figure 5B and Appendix A). However, none were associated with a single H-L pair (2 had multiple heavy, 1 had multiple light chains), which may indicate a lack of function or dysfunctional development of IgK positive B cells. From the 10x experiment, IgK accounted for only 0.5% of the light chains. This low IgK frequency may be due to a suboptimal constant region reverse primer being used (IgKC2 used in the nested PCR of the 10x experiment versus IgKC1 used in mid-throughput strategy) or a difference between individual animals.

The combined analysis of antibody light chains from the three animals revealed that 39.1% (nine chains, out of which seven were distinct), 64.5% (20; 17 distinct), and 15.1% (720; 21 distinct) of the light chains from strategies 1, 2, and 3 were shared between at least two individuals (public chains, Appendix A). Out of the distinct chains, three were shared between all three animals. No public heavy chains were observed, consistent with previous reports [36]. Altogether, increasing the throughput of antibody discovery in cattle could shed light on the conflicting issue of heavy and light chain pairing and sharing. In particular, identifying public antibodies among vaccinated or infected animals and convergent CDRH3 could allow the characterization of the antigen-specific antibody response [80].

## 4. Conclusions

All three workflows successfully isolated natural H-L antibody pairs from single cattle B cells, with no obvious bias between the different approaches, despite using three different animals. This is an advance to enable the study of the antibody response that in turn may inform the optimization or development of new vaccines. The three throughput methods described here have pros and cons when selecting the most suitable strategy for a given aim (Table 2). However, both the single-cell amplification methods and the 10x workflow are very sensitive to cell quality, with the former requiring intact heavy and light chain transcripts and the latter being sensitive to ambient RNA contamination from non-intact cells.

When considering the most appropriate method, it is essential to determine the number of H-L pairs required. To equal the throughput of strategy 3, hundreds of plates using either strategy 1 or 2 would need to be processed, with correspondingly high expenditure. If the B cell isolation includes antigen specificity, a few cells may be enough instead of analyzing thousands of cells with feature barcodes to identify the rare population of antigen-specific cells in the high-throughput strategy. The time frame should also be considered since the cloning or Sanger sequencing approach, although robust, is time-consuming and PCR or NGS is more time-efficient. Another obvious decision-making factor is the availability of specialist equipment such as a 10x device, although equivalent instrument-free approaches are likely to emerge [91]. Alternatively, the intermediate-throughput approach enables several hundred antibodies to be sequenced in one sequencing run. Importantly, the flow cytometry approaches are independent of downstream processing, and any PCR strategy (i.e., multiplexed RT-PCR *versus* singleplex PCR including IgK chains) can interchangeably be Sanger-sequenced or indexed for NGS.

The other key consideration linked to the throughput is the cost. A guide to reagent and consumable costing for each method amplification or sequencing module is provided based on the experiments in this study (Table 2). In brief, the low and medium strategies cost approximately $80 USD per sequenced antibody, while the high-throughput approach costs $2 USD per antibody (Table 2). Although the costings for the low-throughput strategy are fixed, pooling antibodies from multiple plates to sequence 384 antibodies simultaneously can considerably decrease the price/antibody for the medium-throughput approach. We also tested desalt indexing primers and found no difference in antibody sequences compared to Illumina HPLC primers. However, the efficiency was slightly reduced, which was compensated for by two additional cycles in the indexing PCR. Using desalt primers for the medium-throughput approach will have an initial cost but would be cost-effective in the long term.

Overall, the three strategies described here enable successful cattle natural antibody H-L pair sequencing. In particular, 10x protocols were adapted to cattle antibody sequencing for the first time. Implementing these strategies with antigen-specific flow cytometry sorting or feature barcoding before 10x GEM generation would enable the antigen-specific component of the antibody response to vaccination or infection to be identified. Finally, this workflow can be applied to sequence antibodies in other veterinary species.

## Figures and Tables

**Figure 1 vaccines-11-01099-f001:**
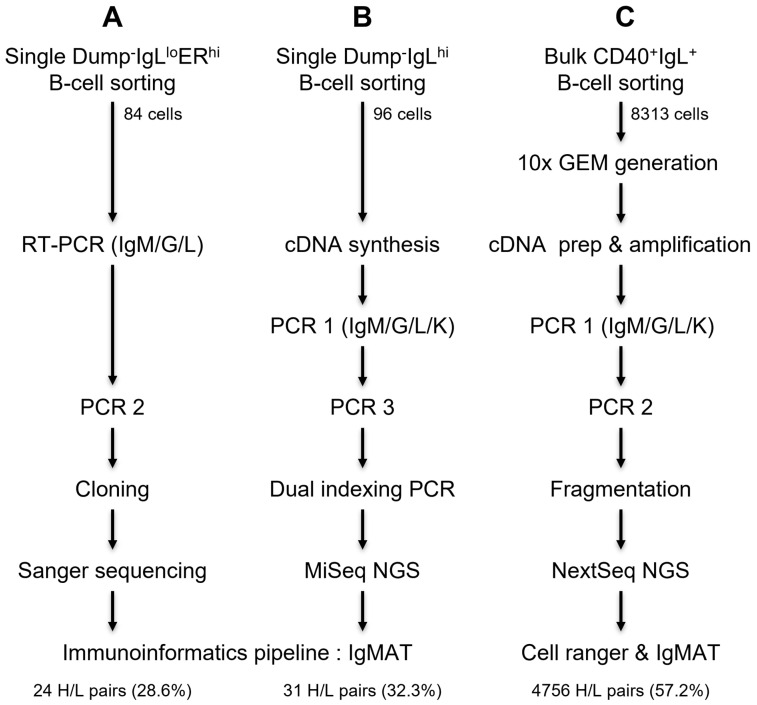
Three strategies for antibody heavy- and light-chain natural pair sequencing. B cells are single-cell-sorted in plates, and a semi-nested PCR is followed by cloning and Sanger sequencing ((**A**) strategy 1—low-throughput workflow) or Illumina adaptors and indexes are added by PCR amplification followed by NGS ((**B**) strategy 2—medium-throughput workflow). B cells are bulk-sorted and single cells are partitioned in Gel Beads-in-Emulsion (GEM) using 10x Chromium Controller followed by semi-nested PCR, fragmentation, and NGS ((**C**) strategy 3—high-throughput workflow). Analyses of sequenced heavy and light chains are performed using the Ig-Sequence Multi-Species Annotation Tool (IgMAT, [51]).

**Figure 2 vaccines-11-01099-f002:**
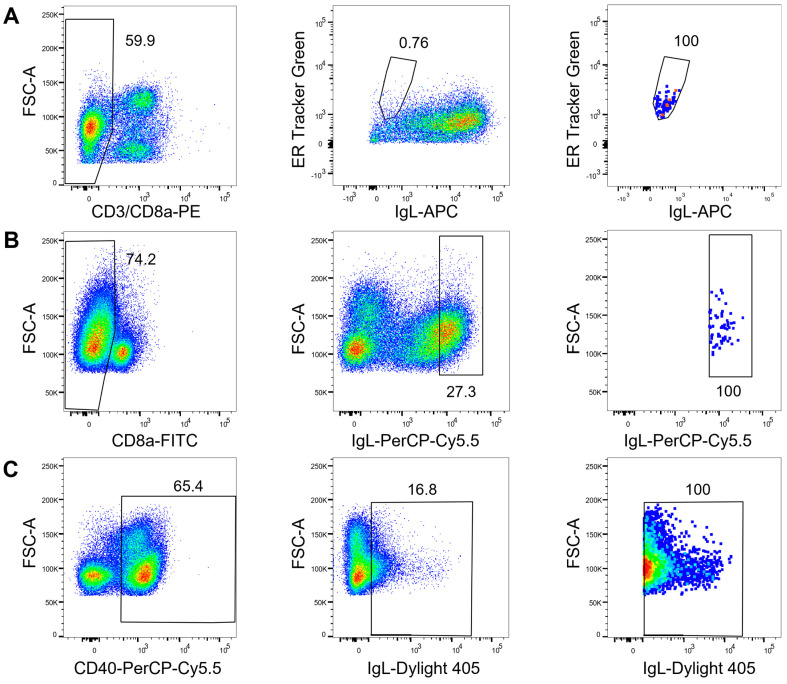
Three strategies of flow cytometry B cell gating. Plots in the left column represent enrichment gating strategies, the middle column represents the sort gating strategy, and the right column shows the sorted cells: (**A**) CD3^-^CD8a^-^IgL ^low^ER^hi^ cells were single-cell-sorted for strategy 1; (**B**) CD8a^-^IgL^hi^ single cells were sorted for strategy 2; (**C**) CD40^+^IgL^+^ cells were sorted in bulk for strategy 3. All gatings were performed on live singlets lymphocytes. Percentages within each gate are indicated on the plots.

**Figure 3 vaccines-11-01099-f003:**
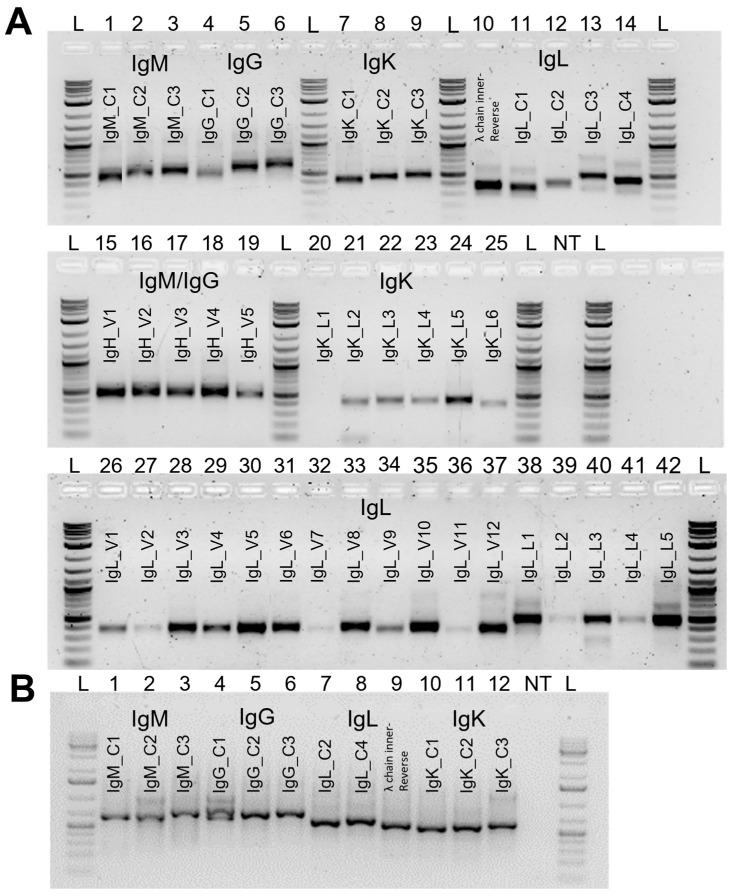
PCR primer validation. (**A**) Cattle cDNA amplification using heavy and light antibody chain primers. The first row shows amplicons generated using the set of constant region primers from the heavy IgM (1–3: IgM_C1-3) and IgG (4–6: IgG_C1-3), and the light IgK (7–9: IgK_C1-C3) and IgL (10: Ig λ chain inner-reverse [31], 11–14: IgL_C1-4) chains paired with IgH_V4 (1–6), IgK_L3 (7–9), and IgL_V6 (10–12, 14) or IgL_L1 (13), respectively. The second row shows amplicons from individual heavy-chain VHset primers paired with IgM_C2 (15–19), and the kappa chain individual VKset primers paired to IgK_C2 (20–25). The third row shows individual lambda light chains primers from the VL12set (26–37) paired with IgL_C2 and the individual VL5set primers (38–42) paired to IgL_C4. All amplicons are between 400 bp and 600 bp. (**B**) The 10x barcoded cDNA amplification using Read1 forward and cattle constant region reverse primers. The Read1 forward primer [57] was paired with several reverse primers to select suitable pairs for IgM (1–3: IgM_C1-3), IgG (4–6: IgG_C1-3), IgL (7: IgL_C2; 8: IgL_C4; 9: Ig λ chain inner-reverse [31]) and IgK (10–12: IgK_C1-3). NT: no template control; L: NEB 1 Kb Plus ladder.

**Figure 4 vaccines-11-01099-f004:**
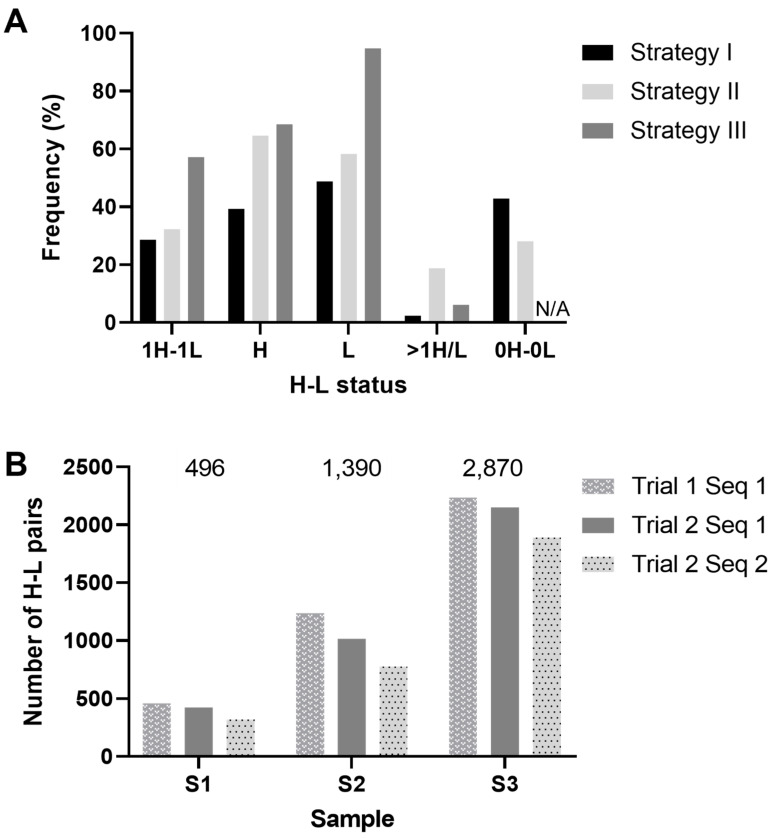
Proportions of sequenced heavy (H) and light (L) chains. (**A**) Frequencies of H/L chain statuses for the three strategies. Percentages of each status are indicated as frequencies of wells processed (strategies 1 and 2) or estimated numbers of cells (strategy 3): 1H-1L: single H and single L pairs; H: overall H chains amplified (including multiplets); L: overall L chains amplified (inc. multiplets); >1H/L: wells or 10x GEMs with more than one H or L chain; 0H-0L: no amplification of either chain. (**B**) Within strategy 3, the numbers of antibody pairs recovered for sample 1 (S1), sample 2 (S2), and sample 3 (S3) are shown. The pairs of single H and single L chains obtained for the two trials (trial 1 and 2) and two sequencing runs (Seq 1 and Seq 2) are shown. Total numbers of antibodies recovered for each sample are indicated above each sample’s set of bars.

**Figure 5 vaccines-11-01099-f005:**
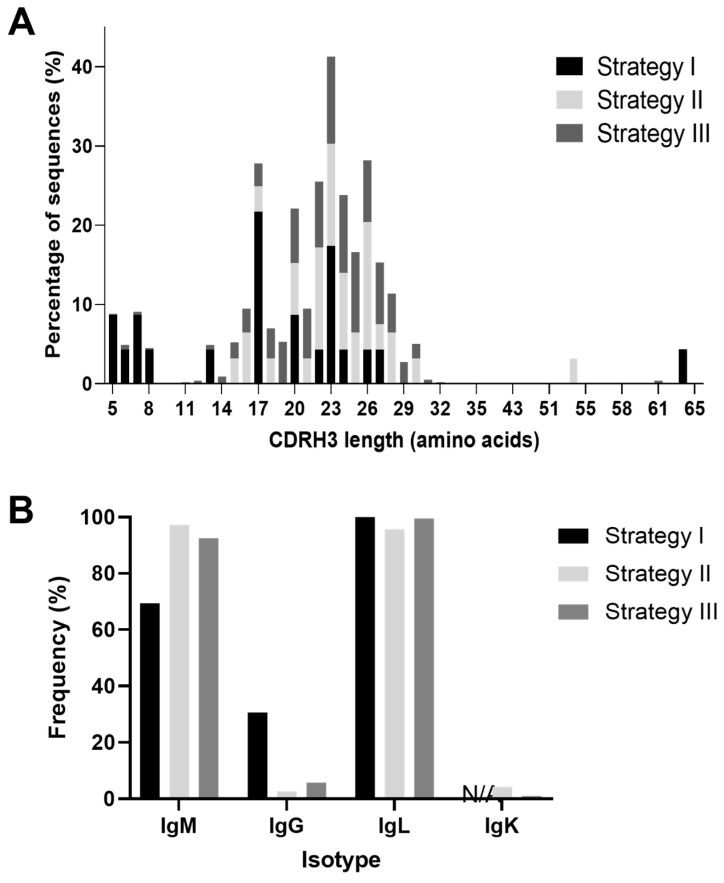
Antibody characteristics. (**A**) Heavy chain complementarity determining region 3 (CDRH3) length distribution from antibodies recovered using the three strategies. Percentages of sequences for each CDRH3 amino acid length are shown. (**B**) H- and L-chain isotypes from all productive chains sequenced by the 3 strategies. Frequencies of heavy-chain IgM and IgG and light-chain lambda (IgL) and kappa (IgK) are shown. IgK frequency was not assessed using strategy 1 (N/A).

**Table 1 vaccines-11-01099-t001:** Cattle heavy- and light-chain targeted primer sequences used for the different amplification strategies *.

Primer Name	Primer Sequence	Orientation	Workflow 1		Workflow 2		Workflow 3	
			RT-PCR	PCR 2	PCR 1	PCR 3	PCR 1	PCR 2
IgG_C3 ^#^	GGCACCCGAGTTCCAGGTCA	RV	✓		✓		✓	
IgG_C2	CACCGGCTCGGGCATGTAGCTGG	RV						✓
IgG_C1	GGGTAGACTTTCGGGGCTGTGGTGG	RV		✓				
IgG_C1_OH	GTCTCGTGGGCTCGGAGATGTGTATAAGAGACAGGTAGACTTTCGGGGCTGTGGT	RV				✓		
IgM_C3	AGGGCCACCGTGCTCTCATC	RV			✓		✓	
IgM_C2	CGAGCTCACGCAGGACACCA	RV	✓					✓
IgM_C1	CTTTCGGGTGTGATTCACCT	RV		✓				
IgM_C2_OH	GTCTCGTGGGCTCGGAGATGTGTATAAGAGACAGGACTCTCRGGTGTGATTCAC	RV				✓		
IgM1_C1_OH	GTCTCGTGGGCTCGGAGATGTGTATAAGAGACAGGACTCTCGGGAGCGATTCAC	RV				✓		
IgH_V1	ACTGTGGACCCTCCTCYTKGTGY	FW	✓	✓	✓			
IgH_V2	ACTGTGGACCCTSSTCTTTGTGC	FW	✓	✓	✓			
IgH_V3	ACTGTGGRCYCTCCTCTTTGTGC	FW	✓	✓	✓			
IgH_V4	VYYGTGGACCCTCCTCTTTGTGC	FW	✓	✓	✓			
IgH_V5	ACTGTGGACCCTCCTCTTTVTVC	FW	✓	✓	✓			
IgH_V1_OH	TCGTCGGCAGCGTCAGATGTGTATAAGAGACAGACTGTGGACCCTCCTCYTKGTGY	FW				✓		
IgH_V2_OH	TCGTCGGCAGCGTCAGATGTGTATAAGAGACAGACTGTGGACCCTSSTCTTTGTGC	FW				✓		
IgH_V3_OH	TCGTCGGCAGCGTCAGATGTGTATAAGAGACAGACTGTGGRCYCTCCTCTTTGTGC	FW				✓		
IgH_V4_OH	TCGTCGGCAGCGTCAGATGTGTATAAGAGACAGVYYGTGGACCCTCCTCTTTGTGC	FW				✓		
IgH_V5_OH	TCGTCGGCAGCGTCAGATGTGTATAAGAGACAGACTGTGGACCCTCCTCTTTVTVC	FW				✓		
IgL_C4	CGGGTAGAAGTCGCTGATGA	RV			✓		✓	
IgL_C2	CCGTTGAGCTCCTCCGTGGAG	RV	✓					
IgL_C1	CGAGGGTGSGGACTTGGGCTGAC	RV		✓				
Igλ chain Inner-Reverse ^#^	GCGGGAACAGGGTGACCGAG	RV						✓
IgL_C3_OH	GTCTCGTGGGCTCGGAGATGTGTATAAGAGACAGCGGGTAGAAGTCGCTGATGA	RV				✓		
IgL_V1	CAGGCTSYACTGACTCAGCCR	FW	✓	✓				
IgL_V2	CAGVCTGKSCTGACTCAGCCK	FW	✓	✓				
IgL_V3	CAGGMTRTGCTGACKCAGCCG	FW	✓	✓				
IgL_V4	CAGGMTVTRCTGACTCAGCCG	FW	✓	✓				
IgL_V5	CAGGMTRTGCTGACKCAGCCG	FW	✓	✓				
IgL_V6	CAGGCTGGYCTGACTCAGCCG	FW	✓	✓				
IgL_V7	CAGGCTGTVCTRACBCAGCCG	FW	✓	✓				
IgL_V8	CAGGCTGTGCTKRCTCARCCG	FW	✓	✓				
IgL_V9	CAGGCTGTKYTGACTCAGCCR	FW	✓	✓				
IgL_V10	CAGGGTGTGCTGACTCAGCCR	FW	✓	✓				
IgL_V11	TCSTATGAACTGACMCAGYYG	FW	✓	✓				
IgL_V12	TCTTCTCARCTGACTCAGCCG	FW	✓	✓				
IgL_L1	ATGGCCYGGTCCCCTCTG	FW			✓			
IgL_L2	ATGGCCTTGGCCCCTCTG	FW			✓			
IgL_L3	ATGGCCTGGATGCTGCTT	FW			✓			
IgL_L4	ATGGYCTGGGCTCTGCTY	FW			✓			
IgL_L5	ATGGCCTGGACCCCTCTC	FW			✓			
IgL_L1_OH	TCGTCGGCAGCGTCAGATGTGTATAAGAGACAGATGGCCYGGTCCCCTCTG	FW				✓		
IgL_L2_OH	TCGTCGGCAGCGTCAGATGTGTATAAGAGACAGATGGCCTTGGCCCCTCTG	FW				✓		
IgL_L3_OH	TCGTCGGCAGCGTCAGATGTGTATAAGAGACAGATGGCCTGGATGCTGCTT	FW				✓		
IgL_L4_OH	TCGTCGGCAGCGTCAGATGTGTATAAGAGACAGATGGYCTGGGCTCTGCTY	FW				✓		
IgL_L5_OH	TCGTCGGCAGCGTCAGATGTGTATAAGAGACAGATGGCCTGGACCCCTCTC	FW				✓		
IgK_C3	TTCACCAAGCACACGACAGA	RV			✓		✓	
IgK_C2	TTCAGCTGCTCATCAGATGGTT	RV						✓
IgK_C1_OH	GTCTCGTGGGCTCGGAGATGTGTATAAGAGACAGGAAGACGGATGGCTCAGCATCAGACC	RV				✓		
IgK_L1	CCTTGGTCTCCTGCTGCTC	FW						
IgK_L2	CCCACTCAGCTCCTCAGTCT	FW						
IgK_L3	TGAGATTCYCTGCTCAGYTCC	FW			✓			
IgK_L4	ATGAGGTTCCCTGTCAGCTC	FW			✓			
IgK_L5	ARATTCCCTGCTCAGCTCCT	FW			✓			
IgK_L6	CTGTTCTTCTGGCTCCCAGC	FW			✓			
IgK_L3_OH	TCGTCGGCAGCGTCAGATGTGTATAAGAGACAGTGAGATTCYCTGCTCAGYTCC	FW				✓		
IgK_L4_OH	TCGTCGGCAGCGTCAGATGTGTATAAGAGACAGATGAGGTTCCCTGTCAGCTC	FW				✓		
IgK_L5_OH	TCGTCGGCAGCGTCAGATGTGTATAAGAGACAGARATTCCCTGCTCAGCTCCT	FW				✓		
IgK_L6_OH	TCGTCGGCAGCGTCAGATGTGTATAAGAGACAGCTGTTCTTCTGGCTCCCAGC	FW				✓		
Read1 primer ^#^	TCTACACTCTTTCCCTACACGACG	FW					✓	
P5+ Read1	AATGATACGGCGACCACCGAGATCTACACTCTTTCCCTACACGAC	FW						✓

* Gene-specific primers designed for amplifying the heavy and light chains of cattle antibodies by using three different throughput approaches; Ticks identify primers used in each strategy. ^#^ Primer sequences reported elsewhere: IgG_C3 [15], Igλ chain inner-reverse [31], Read1 primer [57]; other primers were designed in- house.

**Table 2 vaccines-11-01099-t002:** Features of the three workflows used in the study.

		Workflow 1	Workflow 2	Workflow 3
Flow cytometry	Staining strategy	Stain PBMCs with fluorescently labeled B cell markers		
	Sorting	Single-cell in plates	Single-cell in plates	Bulk sorting in tubes
	Cost differences	Cost of staining and sorting is equivalent for the 3 strategies		
	Duration	1 day	1 day	1 day
Amplification	cDNA preparation	RT-PCR	cDNA preparation	10x + cDNA preparation + amplification
	PCR amplification	semi-nested PCR	semi-nested PCR	semi-nested PCR
	Purification	Gel purification	AMPure bead purification	AMPure bead purification
	Sample preparation for sequencing	pGEM-T cloning + test PCR	PCR-indexing	Fragmentation + indexing
	Duration	10–12 days	3–4 days	2–3 days
	Cost of reagents *	1750 USD/plate	875 USD/plate	1250 USD/sample
Sequencing	Platform	Sanger	Illumina	Illumina
	Analysis	V-Quest/IgMat	IgMat	Cell Ranger + IgMat
Price/antibody * (excluding flow cytometry and FTE)		80 USD	80 USD	2 USD

* Prices calculated on reagents and sequencing costs only (no FTE) for the experiment detailed in the current study.

## Data Availability

The data presented in this study are available as BioProject accession PRJNA962094.

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
