# Peer review of "A Customizable Suite of Methods to Sequence and Annotate Cattle Antibodies"

_vaccines, 2023, doi:10.3390/vaccines11061099_

Round 1
Reviewer 1 Report
The mananuscript of “A customisable suite of methods to sequence and annotate cattle antibodies
” described a high throughput antibody sequencing technology, three strategies were applied in this study by using flow cytometry, single cell sorting, heavy and light chain amplification and sequencing in cattle. In this study, totally 8000 cattle B cells were processed, and 4756 antibody heavy-light chain pairs were sequenced.Overall, this study provided a high throughput antibody sequencing technology, which is important for vaccine and monoclonal antibodies development.
Comments
1 Detailed information of the cattle should be provided, such as the vaccination and infection history.
2. The catalog of the antibodies used in this study should be included in the draft.
Author Response
Thank you for your comments, please find below a point by point response:
1. Detailed information of the cattle should be provided, such as the vaccination and infection history.
All the samples were taken from a high-health herd from animals that showed no overt signs of infection at the time of sampling. The animals of the three strategies were vaccinated using a standard regime for dairy cattle in the UK, and that offers protection against Bovine Respiratory Syncytial virus and other respiratory viruses (Bovipast, Rispoval), lungworm (Huskvac), clostridial diseases (Bravoxin), and Leptospirosis (Spirovac). It should be noted that there was no antigen specific component to the B cells isolation in this study.
The following text has been added to the methods “The three animals were vaccinated using a standard regime for dairy cattle in the UK to protect them against leptospirosis, respiratory and clostridial diseases.”
Changes to the text have been highlighted in yellow.
- The catalogue of the antibodies used in this study should be included in the draft.
Thank you for the suggestion. We have added “Table S1: Markers used for staining cattle B cells in the three strategies” included in supplementary tables, indicating antibody clone when applicable, vendor and catalogue number. The numbering of supplemental tables in the text has been corrected accordingly.
Reviewer 2 Report
1- Ref. 1 is a review article, please cite the source articles, which mentioned the efficacy of the mammalian humoral response to infectious diseases (Ref 3-6 in the review article)
2- As you mentioned in the introduction, mouse have more similarity to human, the cattel have several unusual characteristics including more amino acid in the CDRH3 regions. Furthermore, the mice are used more often for the antibody production. But you don’t mention anything in the Introduction, explaining why you used cattle and not mice?
3- Mention where you purchased the reagents using the name of the company, city and country (this must be uniform).
4- Before the isolation of the PBMCs, explain how you treated the animals and how long you waiting for the immunoreaction?
5- Why you don’t use further marker such as CD20, CD21, CD71 or CD138 to conform the subsets of the sorted cells?
6- The text contains different format in few positions (ex. Line 168)
Author Response
Thank you for your comments, please find below a point-by-point response, where the main manuscript has been edited, we have indicted this in the response.
1- Ref. 1 is a review article, please cite the source articles, which mentioned the efficacy of the mammalian humoral response to infectious diseases (Ref 3-6 in the review article)
Thank you for the suggestion. We have added corresponding references.
2- As you mentioned in the introduction, mouse have more similarity to human, the cattle have several unusual characteristics including more amino acid in the CDRH3 regions. Furthermore, the mice are used more often for the antibody production. But you don’t mention anything in the Introduction, explaining why you used cattle and not mice?
This manuscript is presenting a suite of methods that are applicable to studying cattle B cell responses, that includes single B cells. The first and and-last paragraphs of the introduction explain that while cattle health is essential for productivity, tools to study cattle diseases and immune responses still need to be developed, and there are species-specific complexities. This is not presenting a new method to develop monoclonal antibodies and have modified the introduction to make this clearer and emphasise the importance of antibody-discovery in cattle.
Changes to the text have been highlighted in yellow.
3- Mention where you purchased the reagents using the name of the company, city and country (this must be uniform).
We have added city and country each time a company is cited.
4- Before the isolation of the PBMCs, explain how you treated the animals and how long you waiting for the immunoreaction?
All the samples were taken from a high-health herd from animals that showed no overt signs of infection at the time of sampling. The animals of the three strategies were vaccinated using a standard regime for dairy cattle in the UK, and that offers protection against Bovine Respiratory Syncytial virus and other respiratory viruses (Bovipast, Rispoval), lungworm (Huskvac), clostridial diseases (Bravoxin), and Leptospirosis (Spirovac). It should be noted that there was no antigen specific component to the B cells isolation in this study.
The following text has been added to the methods “The three animals were vaccinated using a standard regime for dairy cattle in the UK to protect them against leptospirosis, respiratory and clostridial diseases.”
5- Why you don’t use further marker such as CD20, CD21, CD71 or CD138 to conform the subsets of the sorted cells?
Lines 386-388, we indeed suggest that using additional markers would allow to select specific cattle B-cell subsets. As the current article is a proof of principle for the sequencing of antibody heavy and light chains from single cells, we believe that the subset of B cells is not relevant to the methods presented that are broadly applicable.
6- The text contains different format in few positions (ex. Line 168)
Thank you for noticing. We have now corroborated that the format is consistent in the text.
Reviewer 3 Report
Kristel Ramirez Valdez et al.reported three strategies to sequence and annotate cattle antibodies through combination of flow cytometry, single cell sorting, heavy and light chain amplification. These make a lot of sense and will help understand specific antibody responses after vaccination and develop effective vaccines.
The following questions need to be improved:
1. The author describes three strategies to sequence and annotate cattle antibodies in general. Were three Holstein-Friesian cattle in the paper immunized with certain vaccines? How to obtain a vaccine-specific antibody sequence, especially a neutralizing antibody sequence with therapeutic value? It is suggested to add in the discussion.
2. "Line 103 PBMCs from three Holstein-Friesian cattle and one per strategy below" means that only one PBMC of each strategy is used to sort B cells. Is the influence of individual differences considered?
3. Figure 1. 84, 96 and 8313 cattle B cells were processed using three workflows,and 24, 31, and 4756 antibody heavy-light chain pairs were sequenced,respectively. Indeed, the number of B cells treated by each sequencing strategy is different, and the antibody heavy-light chain pairs obtained are significantly different. Given the throughput of sequencing for each strategy, could the first two strategies achieve the efficiency of effective antibody heavy-light chain pairs by increasing the number of B cells treated?
4. Figure 1. Both Strategy B and C performed IgM/G/L/K amplification in PCR amplification. Why did Strategy A not perform IgK amplification?
5. In order to facilitate the readability of Figure 3, it is suggested that different swimming lanes represent different amplicons to be marked directly on the figure.
6. The legend contents of A and B in Figure 4 are different. It is suggested to change the chart in Figure 4 B.
7.Table 2 lacks a title.
8. Further improve the format of your references.
Author Response
Thank you for your comments, please find below a point-by-point response, where the main manuscript has been edited, we have indicted this in the response.
- The author describes three strategies to sequence and annotate cattle antibodies in general. Were three Holstein-Friesian cattle in the paper immunized with certain vaccines?
All the samples were taken from a high-health herd from animals that showed no overt signs of infection at the time of sampling. The animals of the three strategies were vaccinated using a standard regime for dairy cattle in the UK, and that offers protection against Bovine Respiratory Syncytial virus and other respiratory viruses (Bovipast, Rispoval), lungworm (Huskvac), clostridial diseases (Bravoxin), and Leptospirosis (Spirovac). It should be noted that there was no antigen specific component to the B cells isolation in this study.
The following text has been added to the methods “The three animals were vaccinated using a standard regime for dairy cattle in the UK to protect them against leptospirosis, respiratory and clostridial diseases.”
How to obtain a vaccine-specific antibody sequence, especially a neutralizing antibody sequence with therapeutic value?
It is suggested to add in the discussion.
Lines 388-391, we suggest using fluorescently labelled antigens to identify and sort antigen-specific cells. To our knowledge, there is no assay that would allow the selection of cattle B cells expressing neutralising antibodies specifically.
- "Line 103 PBMCs from three Holstein-Friesian cattle and one per strategy below" means that only one PBMC of each strategy is used to sort B cells. Is the influence of individual differences considered?
As the aim of this study was to generate a B cell sequencing method that is broadly applicable to cattle and perhaps other closely related species, we did not aim to study specific B cell responses in individuals or responses to specific pathogens or vaccines. As such, using different individuals and different blood samples highlights the methods are robust and we hope translatable into other laboratories. This is discussed in lines 580-583, 596 and now 617-618.
- Figure 1. 84, 96 and 8313 cattle B cells were processed using three workflows, and 24, 31, and 4756 antibody heavy-light chain pairs were sequenced, respectively. Indeed, the number of B cells treated by each sequencing strategy is different, and the antibody heavy-light chain pairs obtained are significantly different. Given the throughput of sequencing for each strategy, could the first two strategies achieve the efficiency of effective antibody heavy-light chain pairs by increasing the number of B cells treated?
Given average efficiencies of 30% for strategies I and II, we would need processing 160 plates to reach the same number of antibodies sequenced in strategy III, which would be around USD 140,000 and USD 280,000. We therefore suggest considering the number of antibodies required before selecting the most adequate strategy (lines 630-632). We have added some text for clarity, highlighted in yellow.
- Figure 1. Both Strategy B and C performed IgM/G/L/K amplification in PCR amplification. Why did Strategy A not perform IgK amplification?
Adding IgK to the first strategy was not possible as it would inhibit the multiplexed PCR (line 467). Requirement to include IgK should drive the selection of the amplification approach (i.e., singleplex) but could be used with Sanger sequencing if preferred (lines 639-642).
- In order to facilitate the readability of Figure 3, it is suggested that different swimming lanes represent different amplicons to be marked directly on the figure.
Thank you for your suggestion, we have added the name of the primer tested to each lane and the chain that is being tested, we hope this adds clarity.
- The legend contents of A and B in Figure 4 are different. It is suggested to change the chart in Figure 4 B.
Thank you for your suggestion. We have added a pattern to the bars of figure 4B, and we have modified the legend. We hope this has added clarity.
7.Table 2 lacks a title.
Thank you for noticing. The title “Table 2: Features of the three workflows used in the study” has been added.
- Further improve the format of your references.
The format of the refences has been improved by corroborating all the references’ format is consistent and is following the Chicago-MDPI style.